# Social Interaction, Lifestyle, and Depressive Status: Mediators in the Longitudinal Relationship between Cognitive Function and Instrumental Activities of Daily Living Disability among Older Adults

**DOI:** 10.3390/ijerph19074235

**Published:** 2022-04-01

**Authors:** Qiuhong Li, Chao Wu

**Affiliations:** School of Nursing, Peking University, No. 38 Xueyuan Road, Haidian District, Beijing 100191, China; qiuhong_li@pku.edu.cn

**Keywords:** cognitive function, depressive status, instrumental ADL disability, lifestyle, older adults, social interaction

## Abstract

(1) Background: Cognitive decline is associated with instrumental activities of daily living (IADL) disability. Intervention targeting the mediators of this association will provide a path to avoid cognition-related IADL disability. (2) Methods: This study used data of wave 2008 (baseline) and wave 2014 of Chinese Longitudinal Healthy Longevity Surveys. Structural equation modeling was conducted to examine the mediating effect of social interaction, lifestyle (fruit and vegetable intake; exercise habits), and depressive status on the association between four baseline cognitive function dimensions (measured by the Chinese version of the Mini-Mental State Examination) and five (2014) IADL dimensions (visiting neighbors, shopping, preparing meals, washing clothes, and taking public transportation). (3) Results: Among 1976 older adults, 29.1% developed IADL disability 6 years later. The cognition–disability association was completely mediated by social interaction (estimate = −0.095, *p* < 0.001), lifestyle (estimate = −0.086, *p* < 0.001), and depressive status (estimate = −0.017, *p* = 0.003). The mediating effects of social interaction (46.3% variances explained) and lifestyle (42.0% variances explained) were both larger than that of depressive status (8.3% variances explained). (4) Conclusions: The development of interventions aimed at improving social interaction, depression, and lifestyle could be of value to prevent cognition-related IADL disability.

## 1. Introduction

Aging is one of the biggest known risk factors for many human diseases. Although the incidence of disability has decreased due to improvements in living standards and medical care, there is also a contention that the aging of the population and the improvement of survivability will result in frailer older adults surviving with health problems. Thus, the number of disabled older adults may be increasing [1]. Progressive disability has adverse impacts on older adults’ daily lives and is related to long-term economic burden and mortality [2]. Compared with basic activities of daily living (ADLs), IADLs seem to be more complex daily activities requiring a higher level of cognitive function, such as shopping or managing medications. A study suggests that the association between cognitive function and ADLs depends substantially on IADLs [3]. Moreover, hippocampal and cortical gray matter volumes are associated with IADLs, indicating that cognitive decline contributes to the incidence of IADL disability [4], which has been found to predict the onset of dementia in the future [5]. New insights into the relationship linking cognitive function and IADL disability could help reveal new ways to break the vicious circle between cognitive impairments and IADL disability, improve life independence, and increase life expectancy and well-being of older adults.

The Human Capital Model [6] posits that intellectual capital (e.g., executive function, brain structure/function), physical capital (e.g., physical function), social capital (e.g., social network, social support), emotional capital (e.g., depression), financial capital (e.g., income), and individual capital (e.g., participation in sports, activity knowledge and skills) are linked together. Among the factors that can be modified, reduced social interaction has been found to be associated with early signs of cognitive decline [7]. Previous studies showed that individuals with lower cognitive function take part in fewer social activities [8], indicating a possible bidirectional relationship between cognitive function and social interaction. Social interaction is also essential for the prevention of functional disability [9]. Therefore, we hypothesized that social interaction would be a critical mediator in the cognitive function–IADL disability association. In addition, a healthy lifestyle, including regular exercise [10] and a healthy diet [11], is associated with improved cognitive function [12]. Among many kinds of food, fruit and vegetables, with high levels of folate, vitamins, and antioxidants, have been shown to be related to cognitive function [13], and regular consumption of fruits and vegetables contributes to improved physical function [14]. Exercise decreases IADL disability as well [15]. Hence, we hypothesized that lifestyle would also mediate the association between cognitive function and IADL disability. Moreover, previous studies provided evidence that cognitive decline, especially poorer memory at baseline, is associated with worse depressive symptoms at follow-up [16], and older adults with depression are less physically active compared to those without depression [17,18], which increases disuse, limitation in multiple daily functioning domains, and the risk of IADL disability in older adults [19,20]. This might be because depressive symptoms impair fine operational skills, via neuroinflammatory or morphological changes in the brain, involved in executive control and mood processing [20,21]. The studies examining the effect of depressive symptoms on the association between cognitive function and IADL disability showed that depressive symptoms significantly mediated and moderated the association between them [20,21]. However, so far, no study has compared the contribution of depression and other factors in the relationship between cognitive decline and future IADL disability.

Previous studies focused on the effects of social interaction, lifestyle, and depression on cognitive function and/or IADL disability. However, the existing studies have rarely paid attention to multiple mediations of these factors on the association between cognitive function and IADL disability. In terms of finding the most effective intervention target and path to better improve the life independence of older adults, it would be critical to explore and examine whether there are differences in the contribution of the three modifiable factors on the cognition–disability association. The findings may be helpful to develop a coordinated intervention plan for the prevention of cognitive decline-induced IADL disability in older individuals.

Therefore, this study aimed to test the longitudinal relationship between cognitive function and IADL disability in older adults with respect to the three mediators of social interaction, lifestyle, and depression and compare the differences in the mediating effects between these three mediators. We hypothesized that superior cognitive function at baseline was associated with more social interaction, a healthier lifestyle, and a milder depressive status over a 6-year follow-up, resulting in better IADL performance.

## 2. Materials and Methods

### 2.1. Study Population

This study drew on data from the 2008 to 2014 Chinese Longitudinal Healthy Longevity Surveys (CLHLS) [22]. The CLHLS was a nationally representative and public dataset of older adults based on randomly selected counties and cities covering 22 out of 31 provinces in mainland China. The CLHLS was approved by the Biomedical Ethics Review Committee of Peking University (IRB00001052-13074).

In this study, we used data over a 6-year period with 2 waves of assessments to conduct a prospective analysis. Participants were initially recruited in the 2008 survey (*n* = 16,954). After excluding individuals aged <65 years in the 2008 survey (*n* = 391), disability of IADL in the 2008 survey (*n* = 10,418), death, or loss to follow-up in the 2014 survey (*n* = 2791), missing baseline (2008) cognitive function (*n* = 1026), endpoint (2014) IADL (*n* = 171), social interaction (*n* = 15), lifestyle (*n* = 47) information, and other covariables (*n* = 47), a total of 1976 participants were included in this study (Figure 1).

### 2.2. Measurements

#### 2.2.1. Cognitive Function

Cognitive function in the 2008 survey was assessed by the Chinese version of the Mini-Mental State Examination (MMSE) [22]. The measurement of cognitive function in the questionnaire involved five dimensions: orientation, registration, recall, attention and calculation, and language. Orientation was measured by asking the participants to recall the time of day, month, date, season, place, and food names. Registration was measured by asking the participants to repeat three words just heard, and recall was measured by asking the participants to repeat three words learned earlier. The dimensions of registration and recall were integrated into the dimension of episodic memory in this study [23]. Attention and calculation were measured by asking the participants to subtract 3 from 30 serially five times and draw a figure of overlapping pentagons. Language was measured by asking the participants to name the objects, repeat a sentence, and act according to the instructions. Scores ranged from 0 to 12 for orientation, and from 0 to 6 for the other three dimensions (episodic memory, attention and calculation, language). The total score of global cognitive function ranges from 0 to 30 [23], and a higher score indicates superior cognitive function. The Cronbach’s alpha coefficient of MMSE in this study was 0.761.

#### 2.2.2. Instrumental Activities of Daily Living (IADL) Disability

Information on IADL disability drew on the data in the 2014 survey, including visiting neighbors, shopping, preparing meals, washing clothes, and taking public transportation. The participants were asked whether they could complete these tasks by themselves (1 = Yes, I can do it independently; 2 = Yes, but I need some help; 3 = No, I cannot do it) [24]. The total IADL disability score ranged from 5 to 15. The Cronbach’s alpha coefficient of IADL in this study was 0.906. In binary logistic regression analyses, the participants entered the IADL disability group if they reported needing help or inability to perform the task (i.e., a response of 2 or 3 was associated with needing help; a total score of more than 5 was considered IADL disability), for each one of the 5 IADL items. In SEM analyses, the levels of each IADL item (ranging from 1 to 3) were entered into the model. The higher the score of each item, the severer the disability regarding that IADL item.

#### 2.2.3. Social Interaction

Information on social interaction drew on the data in the 2014 survey and was collected by asking the following question: “Do you now regularly perform the following activities? 1. Play cards and/or mahjong; 2. Organized social activities” [25] (5 = almost every day, 4 = once for a week, 3 = once for a month, 2 = sometimes, 1 = never). The total social interaction score ranged from 2 to 10.

#### 2.2.4. Lifestyle

Information on lifestyle drew on the data in the 2014 survey. Lifestyle was constructed based on three factors: daily intake of fruit, daily intake of vegetables, and regular physical activity performance [26]. Information on fruit and vegetables was collected by asking the intake frequency of fruit and vegetables (1 = almost every day, or quite often, 0 = occasionally, or never). Information on regular physical activity was collected by asking “Do you do exercises regularly at present?” (1 = yes, 0 = no). The score of lifestyle rating ranged from 0 to 3.

#### 2.2.5. Depressive Status

Information on the depressive status drew on the data in the 2014 survey and was collected by asking the participants the following questions: “1. Do you always look on the bright side of things? 2. Do you often feel fearful or anxious? 3. Do you often feel lonely and isolated? 4. Do you feel the older you get, the more useless you are? 5. Are you as happy as when you were younger?” [27] (1 = always, 2 = often, 3 = sometimes, 4 = seldom, 5 = never). Question 1 and 5 were reversely scored. The total depressive status score ranged from 5 to 25. The Cronbach’s alpha coefficient of the depressive status in this study was 0.713.

#### 2.2.6. Covariates

As possible confounders, variables that have been demonstrated to be associated with cognitive function and IADL disability in previous studies [23,28], including sociodemographic factors and health-related variables available in the 2008 survey, were examined as covariates in this study. The sociodemographic variables (Table 1) included age, gender, education, regions of residence, marital status, occupation, household income, and medical costs per year. The health-related variables included alcohol consumption, smoking status, self-reported health status, self-reported quality of life, night sleep duration, self-reported sleep quality, body mass index (BMI), and chronic diseases (including self-report hypertension, diabetes, and heart disease).

### 2.3. Statistical Analyses

Statistical analyses were performed using SPSS version 18.0 statistical software (IBM Corp., Armonk, NY, USA). We compared the baseline characteristics between participants with and without endpoint (2014) IADL disability using chi-square tests and *t*-tests. Regression analyses were conducted to test the association among cognitive functions, IADL disability, and the three mediators (social interaction, lifestyle, depressive status) to ensure that these variables were pairwise correlated and that the hypothesis models were statistically separable, which is a prerequisite for mediating analyses with structural equation modeling (SEM).

Covariance-based SEM (CB-SEM) was conducted to explore the mediating role of social interaction, lifestyle, and depressive status in the association between baseline cognitive function and IADL disability over a 6-year follow-up in older adults using AMOS 22.0 (IBM Corp., Armonk, NY, USA). CB-SEM is a multivariate technique to test whether theoretical models are compatible with observed data [29]. In our model, cognitive function was the independent latent variable with four dimensions (orientation, episodic memory, attention/calculation, and language) of MMSE as manifest variables. IADL disability was the dependent endogenous latent variable with five items (visiting neighbors, shopping, preparing meals, washing clothes, and taking public transportation) as its manifest variables. Social interaction (indicated by leisure group activity and organized social activity), lifestyle (indicated by fruit and vegetable intake and exercise habits), and depressive status (manifested by the five indicators including optimistic thinking, fearful/anxious, lonely/isolated, feeling useless, happiness) were three endogenous latent mediators, assessed by each subitem of their measurements. Model fit was assessed using the ratio of χ^2^ and degree of freedom (CMIN/DF), the goodness-of-fit index (GFI), the adjusted goodness-of-fit index (AGFI), the comparative fit index (CFI), the normed fit index (NFI), and the root-mean-square error of approximation (RMSEA). We aimed at an adequate fit: CMIN/DF < 5.0, GFI ≥ 0.90, AGFI ≥ 0.90, CFI ≥ 0.90, NFI ≥ 0.90, RMSEA ≤ 0.08. A bootstrapping (5000 times) method was used to estimate the significance of the indirect effects of the mediators.

## 3. Results

Table 1 shows the baseline (wave 2008) characteristics of the study participants in the presence of IADL disability (wave 2014). A total of 1976 older adults were included in the study (Figure 1). Over a 6-year follow-up of 1976 older adults without IADL disability at baseline, 576 (29.1%) developed IADL disability. There were significant differences between the groups with and without IADL disability in age, gender, education, marital status, occupation, drinking and smoking at present (2008), and BMI (all *p* < 0.01; Table 1). In addition, compared to the participants without IADL disability, those who developed IADL disability were more likely to have less social interaction, a more severe depressive status, a worse lifestyle, and a poorer global and domain-specific cognitive function (all *p* < 0.05; Table 1).

The results of linear regression analyses (covariates included) showed that global cognitive function was significantly associated with the three candidate mediators of social interaction, depressive status, and lifestyle. The results of logistic regression analyses showed that global cognitive function was significantly associated with IADL disability (*p* = 0.023). The associations between IADL disability and social interaction (*p* < 0.001), depressive status (*p* < 0.001), and lifestyle (*p* = 0.017) were significant (on-line Appendix A).

Structural equation modeling was used in this study to analyze and compare the multiple mediating effects of the three potential mediators (social interaction, lifestyle, and depressive status) on the longitudinal association between cognitive function dimensions and IADL disability by obtaining coefficient estimates that reflected the degree of the relationship between the variables, including the direct effect, indirect effect, and total effect (i.e., the sum of the direct effect and the indirect effect). Before the mediator entered the model, the association between baseline cognitive function and IADL 6 years later was −0.19 (*p* < 0.001). After the three mediators entered the model, the direct effect of cognitive function on IADL (estimate = −0.007, 95%CI = [−0.124, −0.147], *p* = 0.885) was not statistically significant (Figure 2), indicating that the association linking cognitive function and IADL disability was completely mediated by social interaction, lifestyle, and depressive status. The fit indices of the SEM model were acceptable (Figure 2).

As shown in Table 2, the total effect of cognitive function on IADL disability was statistically significant, and the indirect effects of cognitive function on IADL disability through social interaction, lifestyle, and depressive status were −0.095 (95% CI = [−0.183, −0.007], *p* < 0.001), −0.086 (95% CI = [−0.168, −0.004], *p* < 0.001), and −0.017 (95% CI = [−0.032, −0.003], *p* = 0.003), respectively. Social interaction explained 46.3% of the variances in the association of cognitive function with IADL disability, lifestyle explained 42.0% of the variances, and depressive status explained 8.3% of the variances. In addition, the difference in the mediating effects between social interaction and depressive status, and that between lifestyle and depressive status were both significant (*p* < 0.001 and *p* = 0.013), and the mediating difference between lifestyle and social interaction was not significant. This suggested that social interaction and lifestyle factors (including vegetable and fruit intake and excise habits) could counteract and improve the effect of depression on the cognitive decline-induced operational disability. Furthermore, attention and calculation had the greatest contribution to the salient variable of cognitive function, followed by episodic memory, language, and orientation. Disability in shopping had the greatest contribution to IADL disability, followed by visiting neighbors, preparing meals, washing clothes, and taking public transportation.

## 4. Discussion

The study examined the association between baseline cognitive dimensions and IADL disability over a 6-year follow-up and the mediating roles of social interaction, depressive status, and lifestyle in this association. The results showed that cognitive function was associated with IADL disability 6 years later among older adults, and the association was completely mediated by social interaction, lifestyle, and depressive status.

In this study, 29.1% of the participants developed IADL disability in 6 years. A prospective study in Italy found that 14.3% of nondisabled participants developed IADL disability over 3 years [30], which is lower than the value found in our study. This might be because the 3-year follow-up period was shorter than that in our study. Kim’s study [31] in Korea also reported a lower incidence (21.6%) of IADL disability over a 10-year follow-up. This might be because, compared to our study, there were younger participants, fewer current smokers, and a higher proportion of good self-reported health status in Kim’s study, elements that have been shown to be significantly associated with improved IADL disability [32].

The finding on the negative association between cognitive function and IADL disability is consistent with those of other longitudinal studies [20,33,34], suggesting that more severe cognitive function was associated with an increased risk of IADL disability over the next few years among older adults without baseline IADL disability. Moreover, a previous study [33] indicated that limitations in executive function and episodic memory were important in predicting a greater risk for future functional decline and might cause difficulties in remembering appointments and shopping items, which contributed the most to the latent variable of IADL disability in our study. Moreover, a 6-year predictor study [35] found that the two behavioral variables of attention and memory were the only independent cognitive predictors of IDAL decline after multivariate analysis. In our study, attention and memory made the largest and second-largest contributions to the latent variables of cognitive function.

The results of this study showed that the cognitive function of older adults positively predicted their social interaction and lifestyle and negatively predicted their depressive status. Meanwhile, social interaction, depressive status, and lifestyle have significant effects on the IADL disability of older adults. This means that cognitive function negatively affected IADL disability by upregulating social interaction and lifestyle, as well as downregulating the depressive status. The endpoint social interaction and lifestyle contributed more indirect effects than the depressive status to the longitudinal association between cognition and IADL disability.

Older adults may receive higher social support and find companions to perform sport activities together through increased social interactions [36]. A review [37] revealed that social support was reliably associated with beneficial effects on the cardiovascular, endocrine, and immune systems. In addition, physical activity has been shown to be related to higher social support and closer friends; thus, those with increased social support may have a greater opportunity to exercise [38]. Moreover, emotional support from others is positively associated with intrinsic motivation for physical activity, suggesting that more emotional support from others will promote pleasure and enjoyment in physical activity, making people feel more motivated to exercise [39]. In this study, lifestyle was constructed based on three factors: daily intake of fruit, daily intake of vegetables, and regular exercise practice; we found positive effects of lifestyle on IADL, which is in line with previous findings that participants with regular exercise habits might be less prone to functional disability than those without the same habits [40], and frequent consumption of fruits and vegetables is beneficial to the physical functions of older adults, including IADL and grip strength [14]. 

On the other hand, social interaction could also mediate the cognitive function–IADL disability association through psychological access. Frequent social interactions may provide various emotional benefits through enjoyable and meaningful experiences, such as a sense of belonging and companionship [41], therefore improving both physical and mental health [42]. Second, social interactions may involve the performance of social roles, which promotes a sense of meaning and self-efficacy [43]. Low self-efficacy has been found to be predictive of functional decline among older adults [44]. A study [45] performed SEM on IADLs in China and found that social interactions such as hanging out and playing cards were shown to be significantly associated with IADLs. 

The mechanism underlying the mediating role of depressive status in the cognitive function–IADL disability association may be linked to a lack of interest in all or most things [46]. Older individuals with more severe depressive symptoms reported more negative social interactions [47] and a lack of sense of belonging [48]. A review [49] including 66 studies showed that low social support, social activity restriction, and tense interpersonal relationships are all related to depression. Therefore, people with a depressive status were potentially less engaged in physical activities, which subsequently increased IADL disability [50]. Another explanation is linked to changes in brain function and structure. It has been reported that depression is related to impaired neuroplasticity of the medial prefrontal cortex and hippocampus; moreover, memory deficits have been hypothesized to be directly associated with neuroplasticity deficits in the hippocampus [51]. Previous studies [52] found that cognitive impairments, especially memory deficits, are evident in patients with depression and may reduce individuals’ capacity to finish IADLs, such as shopping and taking transportation. Second, the “vascular hypothesis” of depression proposes that cerebrovascular disease of the frontal lobe leads to emotional changes [53] and is associated with IADL disability [54]. Depression in older individuals was related to prefrontal subcortical circuit lesions, with the most common neuroimaging abnormalities being increased signal hyperintensity in the deep white matter of the frontal lobe and gray matter of the basal ganglia [55]; additionally, depression itself has been strongly associated with coronary and cerebrovascular events [56]. Hence, the mediating effect of depressive status may be linked to brain function and structural changes that lead to IADL disability.

This study has several limitations. First, the depressive status and IADL disability data were based on self-reported measures, which might have resulted in information bias. Second, the assessments of lifestyle, namely, exercise practice and food intake, lacked the specific intakes and duration. Third, a large proportion of data were excluded due to missing information, which might have caused potential biases in sampling. This study’s strength includes the fact that, with a longitudinal study design, the temporal ordering of the association between cognitive function and incident IADL can be validated [20] from the current study. In addition, the relatively large sample size made the results robust, with sufficient statistical power.

The current findings have important implications for clinical practice and intervention. The results of this study contribute to the current knowledge by providing evidence that improving social interaction, lifestyle, and depressive status may alleviate the negative effects of cognitive decline on IADL disability. Specifically, the current study found that, after adjustment for certain other demographic and health-related factors related to cognition and disability, the three influencing factors had a complete mediating effect on the cognition–IADL association, which explained the behavioral mechanism of cognitive decline-induced instrumental disability to a certain extent. Moreover, the current study found that social interaction and lifestyle contributed equally to the longitudinal association between cognitive decline and IADL disability, and their effects were significantly higher than the mediating effect of depression. Furthermore, we found that recreational social activities and organized social activities contributed the same to the social interaction factor in the model. As regards diet, daily vegetable intake, fruit intake, and exercise habits contributed equally to the lifestyle factor in the model; loneliness and anxiety/fear had the highest contribution to depression in the model. These findings provide data support for the formulation of rehabilitation care programs for instrumental disability caused by cognitive decline, that is, for the accurate design of intervention time, duration, and intensity of each factor in the joint intervention plan according to their contributions. Moreover, the period during which an older adult exhibits mild functional limitation but remains independent in IADLs may provide a critical window of opportunity to intervene. Fortunately, these mediators appear to be modifiable, suggesting opportunities for future interventions.

## 5. Conclusions

This retrospective longitudinal study showed that cognitive dimensions are associated with IADL disability in older adults, and the effect of cognitive function on IADL disability is completely mediated by social interaction, lifestyle, and depressive status. Further development of coordinated interventions aimed at keeping IADL independence through improving social interaction, depression, and lifestyle could be of value to improve the mental health of older adults and achieve successful aging.

## Figures and Tables

**Figure 1 ijerph-19-04235-f001:**
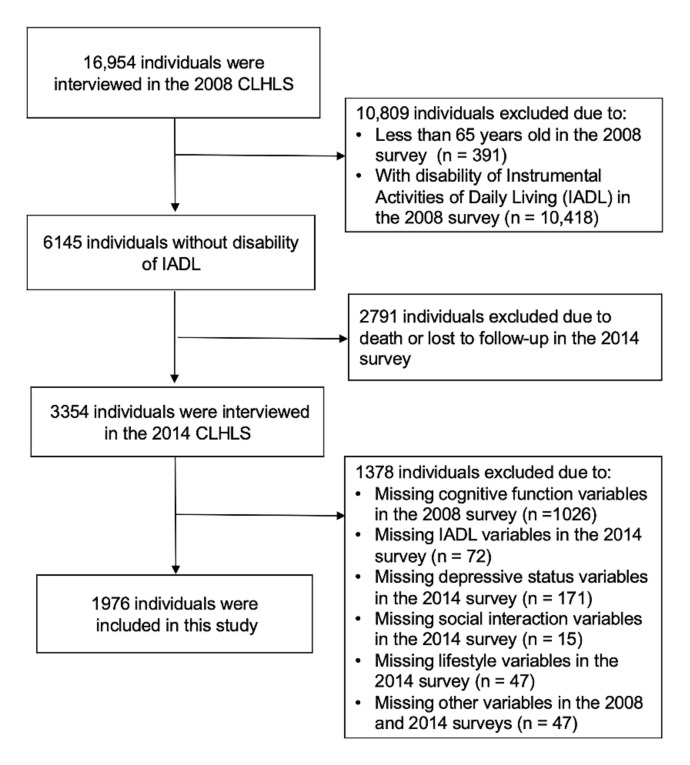
Flowchart of inclusion and exclusion of participants in the study.

**Figure 2 ijerph-19-04235-f002:**
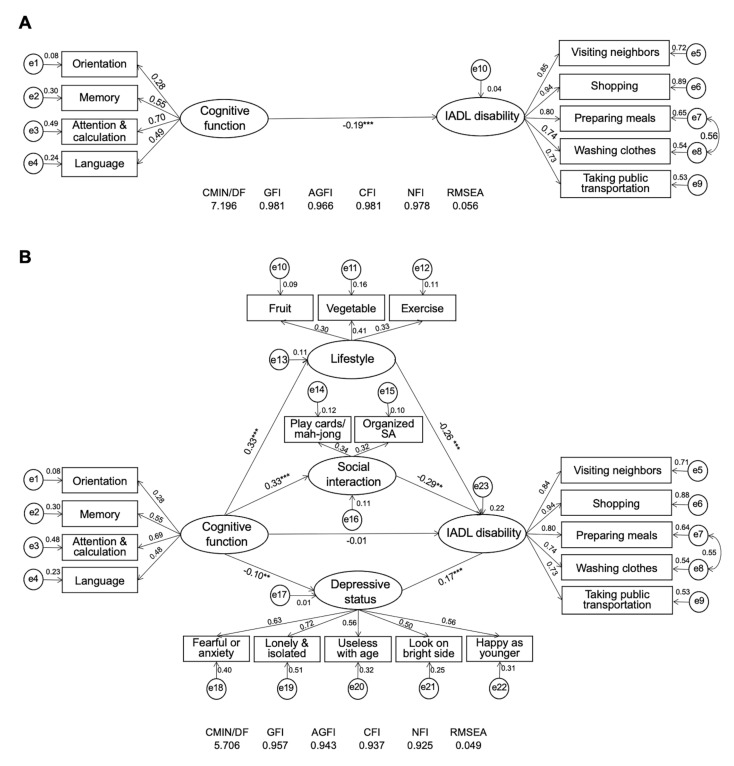
Structural equation modeling of the association between cognitive function and IADL disability. (**A**) Structural equation modeling of cognitive function and IADL disability. (**B**) Structural equation modeling of the mediating effects of social interactions, depressive status, and lifestyle on the cognitive function–IADL disability association. Path coefficients are presented with standardized estimates. ** *p* < 0.01; *** *p* < 0.001. AGFI = adjusted goodness-of-fit index; CFI = comparative fit index; CMIN/DF = ratio of Chi-square and degree of freedom; GFI = goodness-of-fit index; NFI = normed fit index; RMSEA = root-mean-square error of the approximation; SA, social activity.

**Table 1 ijerph-19-04235-t001:** Participants’ baseline (2008) characteristics according to IADL disability in 2014, from Chinese Longitudinal Healthy Longevity Surveys (CLHLS).

Characteristic	Total	Without IADL Disability	With IADL Disability	Statistics	
	(*n* = 1976)	(*n* = 1400)	(*n* = 576)	*t*/χ^2^	*p*-Value
Sociodemographic Variables					
Age, year (SD)	74.3 (7.4)	72.4 (6.2)	79.0 (7.9)	−17.896	<0.001
Gender, *n* (%)				20.212	<0.001
Male	1225 (62.0)	912 (46.2)	313 (15.8)		
Female	751 (38.2)	488 (24.7)	263 (13.3)		
Education, year (SD)	4.0 (4.0)	4.5 (4.1)	2.9 (3.5)	8.630	<0.001
Region of residence, *n* (%)				0.039	0.844
Urban	255 (12.9)	182 (9.2)	73 (3.7)		
Rural	1721 (87.1)	1218 (61.6)	503 (25.5)		
Marital status, *n* (%)				49.416	<0.001
Married and living with a spouse	1295 (65.5)	985 (49.8)	310 (15.7)		
Others	681 (34.5)	415 (21.0)	266 (13.5)		
Occupation, *n* (%)				9.346	0.009
Manual worker	1633 (82.6)	1136 (57.5)	497 (25.2)		
Nonmanual worker	295 (14.9)	231 (11.7)	64 (3.2)		
Others	48 (2.4)	33 (1.7)	15 (0.8)		
HI/year, thousand yuan (SD)	19.5 (22.2)	19.4 (21.7)	19.6 (23.5)	−0.169	0.866
MC/year, thousand yuan (SD)	3.4 (13.5)	3.4 (13.8)	3.2 (12.7)	0.341	0.733
Social interaction (SD)	3.3 (1.9)	3.6 (2.0)	2.6 (1.3)	12.141	<0.001
Play cards and/or mah-jong	1.8 (1.4)	2.0 (1.5)	1.4 (1.0)	9.999	<0.001
Organized social activities	1.5 (1.1)	1.6 (1.2)	1.3 (0.8)	7.595	<0.001
Health-Related Variables					
Drinking at present (2008), *n* (%)	545 (27.6)	417 (21.1)	128 (6.5)	11.688	0.001
Smoking at present (2008), *n* (%)	572 (28.9)	443 (22.4)	129 (6.5)	16.966	<0.001
Exercise at present (2008), *n* (%)	891 (45.1)	628 (31.8)	263 (13.3)	0.106	0.745
Self-reported Health, *n* (%)				0.171	0.918
Good	1177 (59.6)	838 (42.4)	339 (17.2)		
Fair	607 (30.7)	427 (21.6)	180 (9.1)		
Poor	192 (9.7)	135 (6.8)	57 (2.9)		
Self-reported quality of life, *n* (%)				5.545	0.063
Good	1200 (60.7)	831 (42.1)	369 (18.7)		
Fair	697 (35.3)	506 (25.6)	191 (9.7)		
Poor	79 (4.0)	63 (3.2)	16 (0.8)		
Night sleep duration, hour (SD)	7.6 (1.9)	7.5 (1.8)	7.7 (1.9)	−1.866	0.062
Self-reported sleep quality, *n* (%)				0.665	0.717
Good	1389 (70.3)	978 (49.5)	411 (20.8)		
Fair	393 (19.9)	285 (14.4)	108 (5.5)		
Poor	194 (9.8)	137 (6.9)	57 (2.9)		
Body mass index, kg/m^2^ (SD)	21.8 (3.6)	21.9 (3.6)	21.5 (3.5)	2.706	0.007
Chronic diseases, *n* (%)					
Hypertension	457 (23.1)	308 (15.6)	149 (7.5)	3.434	0.064
Diabetes	66 (3.3)	50 (2.5)	16 (0.8)	0.796	0.372
Heart disease	208 (10.5)	144 (7.3)	64 (3.2)	0.295	0.587
Global cognitive function (SD)	28.4 (2.3)	28.6 (2.0)	27.8 (2.7)	6.553	<0.001
Orientation	11.6 (1.0)	11.7 (0.9)	11.5 (1.1)	2.859	0.004
Memory	5.5 (1.0)	5.6 (0.9)	5.4 (1.2)	4.288	<0.001
Attention and calculation	5.4 (1.0)	5.5 (0.9)	5.1 (1.2)	7.034	<0.001
Language	5.9 (0.4)	5.9 (0.4)	5.9 (0.4)	2.306	0.021
Depressive status (SD)	10.8 (3.4)	10.3 (3.3)	11.9 (3.5)	−9.690	<0.001
Lifestyle, score (SD)	1.8 (0.8)	1.9 (0.8)	1.7 (0.8)	6.041	<0.001
Fruit, *n* (%)	902 (45.6)	655 (33.1)	247 (12.5)	2.507	0.113
Vegetable, *n* (%)	1838 (93.0)	1221 (67.4)	507 (25.7)	31.229	<0.001
Exercise, *n* (%)	880 (44.5)	680 (34.4)	200 (10.1)	31.688	<0.001

Notes: HI, household income; IADL = instrumental activities of daily living; MC, medical costs; SD = standard deviation.

**Table 2 ijerph-19-04235-t002:** Mediators in the longitudinal association between cognitive function and the severity of IADL disability.

Variables	Standardized Estimate	Standard Errors	LLCI	ULCI	*p*	Ratio
Direct effect on IADL						
Cognitive function	−0.007	0.072	−0.124	0.147	0.885	-
Indirect effect						
Social interaction (a1 × b1)	−0.095	0.045	−0.183	−0.007	<0.001	46.34%
Lifestyle (a2 × b2)	−0.086	0.042	−0.168	−0.004	<0.001	41.95%
Depressive status (a3 × b3)	−0.017	0.007	−0.032	−0.003	0.003	8.29%
Total effect	−0.205	0.039	−0.282	−0.128	<0.001	-
Difference of indirect effects						
a1 × b1 − 2 × b2	−0.008	0.055	−0.115	0.098	0.875	-
a1 × b1 − a3 × b3	−0.077	0.043	−0.162	0.007	<0.001	-
a2 × b2 − a3 ×b3	−0.069	0.041	−0.149	0.011	0.013	-

Notes: IADL = instrumental activities of daily living; LLCI = lower limit of 95% confidence interval; ULCI = upper limit of 95% confidence interval.

## Data Availability

The raw data used in this study are freely available from the Chinese Longitudinal Healthy Longevity Survey (CLHLS; https://opendata.pku.edu.cn/dataset.xhtml?persistentId=doi%3A10.18170/DVN/WBO7LK, accessed on 30 June 2020).

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
