# Peer review of "Social Interaction, Lifestyle, and Depressive Status: Mediators in the Longitudinal Relationship between Cognitive Function and Instrumental Activities of Daily Living Disability among Older Adults"

_ijerph, 2022, doi:10.3390/ijerph19074235_

Round 1
Reviewer 1 Report
For lay readers of statistics, the figures in Table 2 are difficult to grasp their relevance and the eyes are directed only to the p-value column.
Author Response
Response to Reviewer 1
We gratefully thank the reviewers and editors for the comments that help improve the manuscript. Below the comments (in bold) are our responses point by point, and the manuscript has been revised accordingly. The revised text was in italics in this response letter and highlighted accordingly in yellow in the revised manuscript.
For lay readers of statistics, the figures in Table 2 are difficult to grasp their relevance and the eyes are directed only to the p-value column.
Responses: Thank you for pointing out this problem in our manuscript. According to your suggestion, we have modified the format of Table 2 in the revised manuscript to make the results clearer to read and explained the results in detail in the Results section (lines 218-240),
“Structural equation modeling was used in this study to analyze and compare the multiple mediating effects of the three potential mediators (social interaction, lifestyle, depressive status) on the longitudinal association between cognitive function dimensions and IADL disability by obtaining coefficient estimates that reflect the degree of relationships between variables, including the direct effect, indirect effect, and total effect (i.e., the sum of the direct effect and indirect effect). Before the mediator entered the model, the association between baseline cognitive function and IADL 6 years later was -0.19 (p < 0.001). After the three mediators entered the model, the direct effect of cognitive function on IADL (estimate = -0.007, 95%CI = [-0.124, -0.147], p = 0.885) was not statistically significant (Figure 2), indicating that the association linking cognitive function and IADL disability was completely mediated by social interaction, lifestyle, and depressive status. The fit indices of the SEM model were acceptable (Figure 2).
As shown in Table 2, the total effect of cognitive function on IADL disability was statistically significant, and the indirect effects of cognitive function on IADL disability through social interaction, lifestyle, and depressive status were -0.095 (95% CI = [-0.183, -0.007], p < 0.001), -0.086 (95% CI = [-0.168, -0.004], p < 0.001), and -0.017 (95% CI = [-0.032, -0.003], p = 0.003), respectively. Social interaction explained 46.3% of the variances in the association of cognitive function with IADL disability, lifestyle explained 42.0% of the variances, and depressive status explained 8.3% of the variances. In addition, the difference in the mediating effects between social interaction and depressive status, and that between lifestyle and depressive status, were both significant (p < 0.001 and p = 0.013), and the mediating difference between lifestyle and social interaction was not significant. It suggested that social interaction and lifestyle (including vegetable and fruit intake and excise habits) could counteract and improve the effect of depression on the cognitive-decline induced operational disability.”

Reviewer 2 Report
This study is a longitudinal study showing that lifestyle mediates cognition-disability associations.
Problem: There is no description of the criteria by which the variables used in this study were selected. Also, there is no description of the model/hypothesis to be tested and previous research it is based on. There is no description of what shortages in previous studies this paper has made up for and what contributions it has made. As a result, I don't know how to evaluate this paper.
Author Response
Response to Reviewer 2
We gratefully thank the reviewers and editors for the comments that help improve the manuscript. Below the comments (in bold) are our responses point by point, and the manuscript has been revised accordingly. The revised text was in italics in this response letter and highlighted accordingly in yellow in the revised manuscript.
There is no description of the criteria by which the variables used in this study were selected.
Response: Thanks for the suggestion. In the revised manuscript, we have added the criteria for selecting the variables used in this study in the Materials and Methods section (lines 162-164) and addressed the hypothesis in the Introduction section (lines 46-71; please also see the responses to the next comment). The main variables including cognitive function, IADL disability, social interaction, lifestyle, and depressive status used in this study were based on the hypothesis model of this study. Covariates were selected based on previous studies.
In Methods (line 159-161),
“As possible confounders, variables that have been demonstrated to be associated with cognitive function and IADL disability in previous studies [23,28], including sociodemographic factors and health-related variables available in the 2008 survey, was examined as covariates in this study. The sociodemographic variables (Table 1) included age, gender, education, regions of residence, marital status, occupation, household income, and medical costs per year. Health-related variables included alcohol consumption, smoking status, self-reported health status, self-reported quality of life, night sleep duration, self-reported sleep quality, body mass index (BMI), and chronic diseases (including self-report hypertension, diabetes, and heart disease).
There is no description of the model/hypothesis to be tested and previous research it is based on. There is no description of what shortages in previous studies this paper has made up for and what contributions it has made.
Response: In the Introduction section, we have addressed the hypothesis to be tested and reviewed the research it is based on in the revised manuscript (lines 74-87).
“Previous studies have focused respectively on the effects of social interaction, lifestyle, and depression on cognitive function and/or IADL disability. However, the existing studies have rarely paid attention to multiple mediations of these factors on the association between cognitive function and IADL disability. In terms of finding the most effective intervention target and path to better improve the life independence of older adults, it would be critical to explore and examine whether there were differences in the path weighting of the three modifiable factors on the cognition-disability association. The findings may be helpful to develop a coordinated intervention plan for the prevention of cognitive-decline induced IADL disability in older individuals.
Therefore, this study aimed to test the longitudinal relationship between cognitive function and IADL disability in older adults through three mediators of social interaction, lifestyle, and depression, and compare the differences in the mediating effects between these three mediators. We hypothesized that superior cognitive function at baseline was associated with more social interaction, healthier lifestyle, and milder depressive status over a 6-year follow-up, resulting in better IADL performance.”

Reviewer 3 Report
The text is well written, complete and interesting. This reviewer would like the text to better describe the software used for the statistical analysis. And also to comment on whether validation analyzes were carried out on hypotheses such as A-nova, etc. Statistical information would be important and appropriate to ensure that such models are statistically separable. In addition, the results seem adequate and the methodology as well.Author Response
Response to Reviewer 3
We gratefully thank the reviewers and editors for the comments that help improve the manuscript. Below the comments (in bold) are our responses point by point, and the manuscript has been revised accordingly. The revised text was in italics in this response letter and highlighted accordingly in yellow in the revised manuscript.
This reviewer would like the text to better describe the software used for the statistical analysis.
Response: We have added the software information and the details of the statistical analysis (lines 177-189) in the revised manuscript.
“The covariance-based SEM (CB-SEM) was conducted to explore the mediating role of social interaction, lifestyle, and depressive status in the association between baseline cognitive function and IADL disability over a six-year follow-up in older adults using AMOS 22.0 (IBM Corp., Armonk, NY, USA). CB-SEM is a multivariate technique to test whether theoretical models are compatible with observed data [29]. In our model, cognitive function was the independent latent variable with four dimensions (orientation, episodic memory, attention/calculation, and language) of MMSE as manifest variables. IADL disability was the dependent endogenous latent variable with five items (visiting neighbors, shopping, preparing meals, washing clothes, and taking public transportation) as its manifest variables. Social interaction (indicated by leisure group activity and organized social activity), lifestyle (indicated by the fruit and vegetable intake exercise habit), and depressive status (manifested by the five indicators including optimistic thinking, fearful/anxious, lonely/isolated, feeling useless, happiness) were three endogenous latent mediators, assessed by each subitem of their measurements.”
And also to comment on whether validation analyzes were carried out on hypotheses such as A-nova, etc. Statistical information would be important and appropriate to ensure that such models are statistically separable.
Response: Thanks for the suggestions. To ensure that the hypothesis models are statistically separable, regression analyses were carried out to test each path of the models before performing structural equation modeling. We have added this information in the revised manuscript as follows (lines 171-175),
“Regression analyses were conducted to test the association among cognitive function, IADL disability, and three mediators (social interaction, lifestyle, depressive status) to ensure that these variables were pairwise correlated and that the hypothesis models were statistically separable, which is a prerequisite for mediating analyses with structural equation modeling (SEM).”

Round 2
Reviewer 2 Report
The author has made great improvements. I think the publication decision is fine.